# Correlation between clinical disease activity and interferon-γ autoantibody titers measured by inhibitory ELISA, and inflammatory biomarkers in adult-onset immunodeficiency associated with anti-interferon-γ autoantibodies

**Putthapon Teepapan[1], Apinya Chungcharoenpanich[1], Kanokkarn Pinyopornpanish[1,2], Supa Oncham[1], Prawat Chantharit[3], Porpon Rotjanapan[3], Wannada Laisuan[1]***

1 Division of Allergy Immunology and Rheumatology, Department of Medicine, Faculty of Medicine Ramathibodi Hospital, Mahidol University, Bangkok, Thailand, 2 Division Allergy and Clinical Immunology, Department of Internal Medicine, Faculty of Medicine, Chiang Mai University, Chiang Mai, Thailand, 3 Division of Infectious Diseases, Department of Medicine, Faculty of Medicine Ramathibodi Hospital, Mahidol University, Bangkok, Thailand

* wannada.lais@gmail.com

## Abstract

### Background

Anti-interferon-γ autoantibodies (Anti-IFN-γ AAbs) contribute to immunodeficiency and increase susceptibility to intracellular infections, particularly in adults. Measurement of anti-IFN-γ AAb titers using inhibitory ELISA is a valuable diagnostic tool for adult-onset immunodeficiency. However, the relationship between inhibitory ELISA titers, inflammatory biomarkers, and clinical disease activity remains unclear.

### Methods

This retrospective study analyzed 69 blood samples from 39 patients with detectable anti-IFN-γ AAbs at Ramathibodi Hospital. Data collected included demographics, clinical disease activity, and laboratory biomarkers such as white blood cell (WBC) count, interleukin-6 (IL-6), C-reactive protein (CRP), erythrocyte sedimentation rate (ESR), and inhibitory ELISA titers. Disease activity was categorized as active or in remission based on clinical evaluation and the status of infection.

### Results

The mean patient age was 58.38 ± 8.55 years, and 61.5% were female. Anti-IFN-γ AAb titers had an area under the receiver operating characteristic curve (AUC) of 0.893. A cut-off of 1:50,000 (50% inhibition) yielded 92.90% specificity in determining active disease status. Median titers were significantly higher in active disease (1:100,000, interquartile range [IQR]: 1:10,000–1:100,000) compared with remission

**Data availability statement:** All relevant data are within the paper.

**Funding:** This study received grant from Ramathibodi Hospital, Mahidol university. The funders had no role in study design, data collection and analysis, decision to publish, or preparation of the manuscript.

**Competing interests:** The authors have declared that no competing interests exist.

(1:5,000, IQR: 1:5,000–1:10,000, $p < 0.001$). Combining anti-IFN-γ AAb titers (≥ 1:50,000) with ESR (≥ 45 mm/hr), CRP (≥ 18 mg/dL), WBC (≥ 10,455 cells/µL), and IL-6 (≥ 16 pg/mL) further improved the prediction of disease activity (AUC = 0.903, sensitivity = 88.9%, specificity = 91.7%).

## Conclusions

High anti-IFN-γ AAb titers correlate with active disease, and inhibitory ELISA may aid in disease monitoring. Combining IFN-γ AAbs titers with inflammatory biomarkers may improve predictive accuracy.

## Introduction

Interferon-γ (IFN-γ) is a pivotal cytokine in the host immune response, essential for orchestrating defense mechanisms against intracellular pathogens, including non-tuberculous mycobacteria (NTM), *Salmonella* species, and various viral agents [1–3]. As a central component of the IFN-γ- interleukin-12 (IL-12) pathway axis, IFN-γ is primarily secreted by Th1 cells, activated cytotoxic T lymphocytes and natural killer (NK) cells. It's signaling facilitates the activation of mononuclear phagocytes, enhances antigen eradication, and modulates a pro-inflammatory cascade involving IL-12 and tumor necrosis factor-alpha (TNF-α) [4].

The criticality of the IFN-γ pathway is well-documented in both experimental and clinical settings. In experimental mouse models, a lack of IFN-γ immunity results in increased susceptibility to Mycobacterium tuberculosis, bacterial pathogens such as Listeria monocytogenes, and viral infections. [5–7] In humans, inborn errors of IFN-γ immunity-such as medelian susceptibility to mycobacterial disease (MSMD), reports variants in 21 genes that impair the production of IFN-γ. A lack of IFN-γ cytokine predisposes to mycobacterial, *Salmonella* species, and macrophage-trophic fungal infections. [8–10]

In recent years, anti-cytokine autoantibodies have emerged as a significant mechanism underlying immunodeficiency disorders [11–14]. Notably, the presence of high-affinity, neutralizing anti-interferon-γ autoantibodies (anti-IFN-γ AAbs) have been identified as the hallmark of a distinct clinical entity: anti-interferon-γ autoantibodies syndrome (AIGA) [1,15–17]. Patients with these autoantibodies typically present with disseminated NTM infections (approximately 80% of cases) and other opportunistic infections involving the lymph nodes, bone, lungs, skin, and bloodstream. [1,18–24]. The clinical progression is often refractory, characterized by a fluctuation course of relapse and remission, with recurrent infections rates ranging from 18–80% [19,20,25,26].

A significant challenge in the clinical management of this syndrome is the unpredictable nature of the disease. [19,20] Currently, there is a critical lack of standardized biomarkers to accurately monitor disease activity or predict clinical flare-ups. While the measurement of anti-IFN-γ AAb titers-specifically through inhibitory ELISA-has shown promise in reflecting the neutralizing capacity of the antibodies. [26–28] Previous studies have reported an association between anti-IFN-γ AAb titers and disease activity,

with an area under the receiver operating characteristic curve (AUC) of 0.8795 [29]. Anti-IFN-γ AAbs titers are not routinely available in most clinical laboratories. In clinical practice, physicians often rely on non-specific inflammatory markers, including white blood cell count (WBC), C-reactive protein (CRP), erythrocyte sedimentation rate (ESR), and hemoglobin (Hb) levels to assess infection status. [25,26] However, the correlation between these markers and the specific pathophysiology of Anti-IFN-γ Aabs remained poorly defined. Identifying reliable, accessible biomarkers is essential for optimizing therapeutic interventions, such as the use of rituximab or other immunosuppressive therapies. [26,27,30] This study aimed to explore the correlation between disease activity, inhibitory ELISA titers, and the combined analysis of other inflammatory biomarkers.

## Materials and methods

### Ethical considerations

The study protocol was conducted as a retrospective analysis within an ongoing prospective cohort and was approved by the Ethics Committee of Ramathibodi Hospital, Mahidol University, Thailand (COA. No. MURA2025/462 and COA No. MURA2021/152). The Institutional Review Board waived the requirement for informed consent due to the retrospective nature of the analysis.

### Study design

This was a retrospective observational study conducted at Ramathibodi Hospital, Mahidol university, using data collected between 15 June, 2025 and 15 August, 2025.

### Study population

Participants were identified from the hospital-based anti-IFN-γ Aabs registry, which prospectively enrolls patients diagnosed with adult-onset immunodeficiency associated with anti-IFN-γ Aabs. The prospective cohort was established to enable longitudinal clinical and laboratory monitoring.

All patients in the cohort received standardized care within a multidisciplinary clinical framework involving infectious specialists and immunologists. Patients were routinely evaluated by both teams to develop individualized treatment plans that integrated antimicrobial therapy and immunosuppressive management when indicated. Clinical and laboratory monitoring was performed at regular intervals as part of routine care.

Laboratory evaluations included complete blood count (CBC), erythrocyte sedimentation rate (ESR), C-reactive protein (CRP), liver function tests, serum creatinine (Cr), interleukin-6 (IL-6) levels and Anti-IFN-γ Aabs titers, which were typically assessed every 3–6 months. Anti-IFN-γ Aabs titers were measured in a certified laboratory using QuantiFERON-based inhibitory ELISA assay, as previously described. [31]

### Eligibility criteria

The total prospective cohort comprised 53 patients with Anti-IFN-γ Aabs-associated with immunodeficiency disease.

Inclusion criteria consisted of patients in the cohort who met the predefined criteria for either active disease or complete remission at the time of laboratory assessment.

Exclusion criteria included patients classified as being in partial remission, defined as patients who demonstrated clinical improvement following initiation of treatment but had persistent residual disease manifestations, such as unresolved lymphadenopathy. Fourteen patients were excluded based on this criterion to improve the accuracy and interpretability of the analysis. After exclusion, a total of 39 patients were included in the final analysis.

### Disease activity classification

Disease status was classified at each follow-up visit as either active disease or remission based on clinical evaluation and physical examination performed by an infectious disease specialist and an immunologist.

Active disease was defined as the presence of new-onset fever, reactive dermatosis, or other clinical signs suggestive of ongoing or recurrent infection.

Disease remission was defined as complete clinical resolution of previously diagnosed infections with no evidence of active disease and antibiotic free.

## Data collection

Demographic, clinical, and laboratory data were retrieved from electronic medical records between 15 June 2025 and 15 August 2025. Laboratory data were analyzed at a single timepoint per sample. Each laboratory samples was categorized as representing either the active disease or remission phase according to the predefined disease activity criteria.

## Statistical analysis

All statistical analyses were conducted using IBM SPSS Statistics Version 27 (IBM Corp., Armonk, NY, USA). Categorical variables were analyzed using Fisher's exact test or Pearson's chi-square test, as appropriate. Continuous variables were summarized as mean ± standard deviation, and proportions (%) and $p$-values were calculated using non-missing data. Comparisons between the active disease and remission groups were made using the Mann–Whitney $U$ test. Optimal cut-off values, sensitivity, and specificity were determined using receiver operating characteristic curve analysis and the Youden index. Multiple variable logistic regression was used to integrate multiple biomarkers into a single predictive model and to evaluate whether their combined inclusion improved discriminative performance, as assessed buy the area under the receiver operating characteristic (ROC) curve (AUC).

## Results

### Baseline characteristics and opportunistic infections

A total of 39 patients with anti-IFN-γ AAbs were enrolled in this study. Among these, 24 (61.5%) were female, with a mean age at enrollment of 58.38 ± 8.55 years (Table 1). A total of 69 blood samples were collected, comprising 41 samples from patients with active disease and 28 from those in remission. Most participants were born in central or northeastern Thailand. Metabolic diseases including diabetic mellitus, hypertension, dyslipidemia and cardiovascular disease were the most common comorbidities, present in 24 patients (61.5%), followed by renal disease in seven patients (17.9%), gastrointestinal disease in six patients (15.4%), and autoimmune or hematologic diseases in five patients each (12.8%).

Lymphadenopathy was the most frequent clinical manifestation, occurring in 34 patients (87.2%), followed by fever in 26 patients (66.7%), and reactive dermatosis in 21 patients (53.8%). Among patients with reactive dermatosis, Sweet's syndrome was the most common subtype, occurring in 18 patients (46.2%), followed by erythema nodosum in four patients (10.3%).

Disseminated NTM infection was the primary opportunistic infection. *Mycobacterium abscessus* was the predominant NTM species, detected in 21 patients (53.8%), followed by *Mycobacterium avium* complex in eight patients (20.5%). *Salmonella* infection was the second most frequent opportunistic infection, reported in 11 patients (28.2%), followed by cryptococcal infection in four patients (10.5%).

### Comparison of laboratory biomarkers between active disease and remission in patients with anti-IFN-γ AAbs

A comparison of laboratory parameters between active disease and remission stages is summarized in Table 2. Significant differences were observed in Hb levels, WBC count, alkaline phosphatase (ALP) levels, inflammatory markers (ESR, CRP, IL-6), and anti-IFN-γ AAb titers. Patients in the active stage had lower Hb levels (median, 10.50 g/dL; interquartile range [IQR], 9.60–12.20]) than those in remission (median, 12.85 g/dL; IQR, 11.70–13.60; $p = 0.001$).

**Table 1. Demographic and clinical characteristics of patients with anti-interferon-γ autoantibodies.**

| Characteristics | n, (%) |
|---|---|
| **Age (years), Mean±SD** | 58.38±8.55 |
| **Sex** | |
| Male | 15 (38.5) |
| Female | 24 (61.5) |
| **Age onset (years), Mean±SD** | 54.89±9.09 |
| **Active disease** | 41 (59.4) |
| **Thailand region of birth** | |
| Central | 18 (46.2) |
| Northeast | 13 (33.3) |
| North | 5 (12.8) |
| South | 2 (5.1) |
| West | 1 (2.6) |
| **Comorbid diseases** | |
| Metabolic diseases | 24 (61.5) |
| Hypertension | 10 (26.3) |
| Dyslipidemia | 8 (20.5) |
| Diabetic mellitus | 4 (10.3) |
| Cardiovascular disease | 2 (5.1) |
| Renal disease | 7 (17.9) |
| Gastrointestinal disease | 6 (15.4) |
| Autoimmune disease | 5 (12.8) |
| Hematologic disease | 5 (12.8) |
| Neurological disease | 3 (7.7) |
| Solid malignancy | 1 (2.6) |
| Other* | 11 (28.2) |
| **Clinical presentation** | |
| Lymphadenopathy | 34 (87.2) |
| Fever | 26 (66.7) |
| Reactive dermatosis | 21 (53.8) |
| Wasting | 19 (48.7) |
| Bone pain | 14 (35.9) |
| Pulmonary infection | 13 (33.3) |
| Skin infection | 7 (17.9) |
| Abdominal pain | 2 (5.1) |
| **Dermatologic manifestation** | |
| Neutrophilic dermatosis | 18 (46.2) |
| Erythema nodosum | 4 (10.3) |
| Pustular | 2 (5.1) |
| Others‡ | 1 (2.5) |
| **Oppurtunistic infection†** | |
| Dissiminated nontuberculous mycobacterial infection | |
| *M.Abscessus* | 21(53.8) |
| *M.avium complex* | 8 (20.5) |
| *M.fortuitum* | 2 (5.1) |
| Other NTM species | 6 (15.4) |
| Salmonella *spp.* | 11 (28.2) |

*(Continued)*

**Table 1.** (Continued)

| Characteristics | n, (%) |
|---|---|
| *Mycobacterium tuberculosis* infection | 5 (12.8) |
| Cryptococcosis | 4 (10.5) |
| Talaromycosis | 1 (2.6) |
| Others | 11 (28.2) |

†Distribution of 69 proven opportunistic pathogens diagnosed during the study period.

‡vasculitis.

*others = thyroid disease, eye disease.

**Table 2. Laboratory biomarkers in patients with anti-interferon-γ autoantibodies: Comparison between active disease and remission.**

| Variables | Active | Remission | p-value |
|---|---|---|---|
| | Median (IQR] | Median (IQR] | |
| CD4 + T cell(%) | 26.75 (20.60 - 30.70) | 21.40 (19.60 - 29.20) | 0.543 |
| CD8 + T cell (%) | 30.80 (27.75 - 40.10) | 26.00 (23.10 - 36.50) | 0.234 |
| NK cell (%) | 29.10 (21.80 - 39.55) | 24.50 (15.40 - 27.60) | 0.138 |
| B cell (%) | 5.70 (2.75 - 8.60) | 10.80 (5.30 - 17.90) | 0.070 |
| Hb (g/dL) | 10.50 (9.60 - 12.20) | 12.85 (11.70 - 13.60) | 0.001* |
| WBC (cumm) | 14,830 (11,070 − 19,670) | 7,380 (5,910 − 8,710) | <0.001* |
| ESR | 65.00 (44.00 - 79.50) | 30.50 (12.50- 50.00) | <0.001* |
| CRP | 61.66 (22.29 - 96.15) | 3.98 (1.16 - 12.08) | <0.001* |
| IL-6 | 17.60 (8.40–38.00) | 4.50 (2.35 - 10.90) | 0.001* |
| ALP | 150 (91 - 270) | 88 (83 - 120) | 0.005* |
| Anti-interferon-γ titer | 1:100,000 (1:10,000–1:100,000) | 1:5,000 (1:5,000–1:10,000) | <0.001* |

*$P < 0.05$.

ALP; alkaline phosphatase, Anti-IFN γ Aabs titers; Anti-interferon-gamma autoantibodies titers, CRP; C-reactive protein, ESR; erythrocyte sedimentary rate, Hb; hemoglobin, NK cell; natural killer cell, WBC; White blood cell count, IL-6; interleukin 6 level.

WBC counts were significantly higher in the active disease (median, 14,830 cells/μL; IQR, 11,070–19,670) compared with remission (7,380 cells/μL; IQR, 5,910–8,710; $p < 0.001$), as were ALP levels (median, 150 U/L; IQR, 91–270 vs. 88 U/L; IQR, 83–120; $p = 0.005$). Inflammatory markers were significantly elevated during active disease, including ESR (median, 65.00 mm/hr; IQR, 44.00–79.50 vs. 30.50 mm/hr; IQR, 12.50–50.00; $p < 0.001$), CRP (median, 61.66 mg/L; IQR, 22.29–96.15 vs. 3.98 mg/L; IQR, 1.16–12.08; $p < 0.001$), and IL-6 (median, 17.60 pg/mL; IQR, 8.40–38.00 vs. 4.50 pg/mL; IQR, 2.35–10.90; $p = 0.001$). Anti-IFN-γ AAb titers were also significantly higher in active disease (median, 1:100,000; IQR, 1:10,000–1:100,000) compared with remission (median, 1:5,000; IQR, 1:5,000–1:10,000; $p < 0.001$). No significant differences were observed in the proportions of CD4[+] T cells, CD8[+] T cells, natural killer cells, or B cells between the two stages.

### Diagnostic performance of biomarkers for differentiating active disease from remission in patients with anti-IFN-γ AAbs

A range of laboratory biomarkers was evaluated for their ability to distinguish active disease from remission in patients with anti-IFN-γ AAbs, summarized in Fig 1 and Table 3.

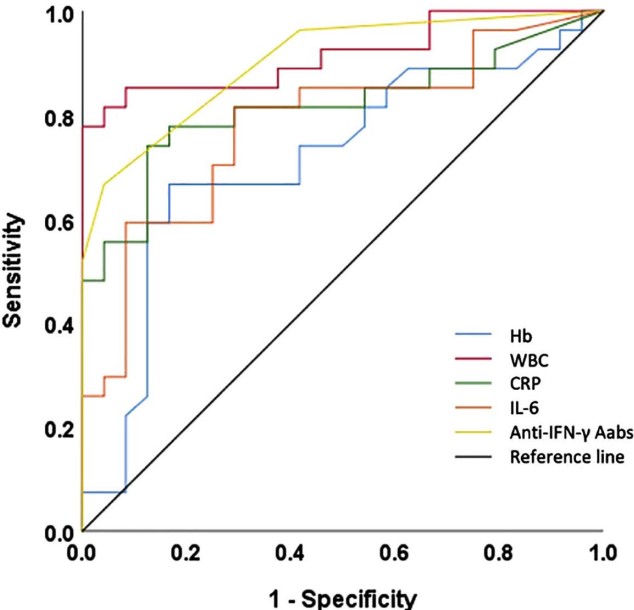

**Fig 1. ROC analysis of laboratory biomarker cut-off values for predicting active disease.**

**Table 3. Diagnostic performance metrics of Anti-IFN-γ autoantibody titers and inflammatory markers.**

| Parameter | AUC (95% CI) | Sensitivity, % (95% CI) | Specificity, % (95% CI) | LR+ (95% CI) | LR− (95% CI) | PPV % (95% CI) | NPV % (95% CI) | Accuracy % (95% CI) |
|---|---|---|---|---|---|---|---|---|
| **Anti–IFN-γ AAbs ≥ 50,000** | 0.893 (0.821–0.965) | 65.9 (49.4–79.9) | 92.9 (76.5–99.1) | 9.22 (2.38–35.7) | 0.37 (0.24–0.57) | 93.1 (77.2–99.2) | 65.0 (48.3–79.4) | 76.8 (65.1–86.1) |
| Anti–IFN-γ AAbs ≥ 10,000 | 0.893 (0.821–0.965) | 97.6 (87.1–99.9) | 53.6 (33.9–72.5) | 2.10 (1.41–3.14) | 0.05 (0.01–0.33) | 75.5 (61.7–86.2) | 93.8 (69.8–99.8) | 79.7 (68.3–88.4) |
| **WBC (≥10,455 cells/μL)** | 0.926 (0.865-0.987) | 82.9 (67.9-92.8) | 92.9 (76.5-99.1) | 11.6 3.03-44.5 | 0.184 (0.0929-0.364) | 94.4 (81.3-99.3) | 78.8 (61.1-91.0) | 87 (77.0-94.0) |
| Hemoglobin (≤11.45 g/dL) | 0.736 (0.614–0.859) | 70.7 (54.5–83.9) | 82.1 (63.1–93.9) | 3.96 (1.75–8.98) | 0.36 (0.22–0.59) | 85.3 (68.9–95.0) | 65.7 (47.8–80.9) | 75.4 (63.5–84.9) |
| **ESR (≥45 mm/hr)** | 0.750 (0.628–0.872) | 75.0 (58.8–87.3) | 75.0 (55.1–89.3) | 3.00 (1.54–5.84) | 0.33 (0.19–0.59) | 81.1 (64.8–92.0) | 67.7 (48.6–83.3) | 75.0 (63.0–84.7) |
| CRP (≥18 mg/L) | 0.864 (0.774–0.954) | 80.0 (64.4–90.9) | 88.9 (70.8–97.6) | 7.20 (2.45–21.2) | 0.23 (0.12–0.42) | 91.4 (76.9–98.2) | 75.0 (56.6–88.5) | 83.6 (72.5–91.5) |
| **IL-6 ((≥16 pg/mL)** | 0.781 (0.652–0.910) | 59.3 (38.8–77.6) | 91.7 (73.0–99.0) | 7.11 (1.82–27.8) | 0.44 (0.28–0.71) | 88.9 (65.3–98.6) | 66.7 (48.2–82.0) | 74.5 (60.4–85.7) |

Anti-IFN γ Aabs titers; Anti-interferon-gamma autoantibodies titers, AUC; area under the curve, CRP; C-reactive protein, ESR; erythrocyte sedimentary rate, Hb; hemoglobin, NPV; negative predictive value, PPV; positive predictive value, LR +; positive likelihood ratio, LR-; negative likelihood ratio, WBC; White blood cell count, IL-6; interleukin 6 level.

## Anti-IFN-γ AAb titers

Anti-IFN-γ AAb titers exhibited strong diagnostic performance, with an AUC of 0.893 at a cut-off ≥ 1:50,000. At this threshold, sensitivity was 65.90% (95% CI 49.4–79.9) and specificity was 92.90% (95% CI 76.5–99.1), yielding a PPV of 93.1% (95% CI 77.2–99.2), an NPV of 65.0% (95% CI 48.3–79.4), and an overall accuracy of 76.80% (95% CI 65.1–86.1). The

LR + was 9.22 (95% CI 2.38–35.7), and the LR − was 0.37 (95% CI 0.24–0.57). A lower cut-off of ≥1:10,000 increased sensitivity to 97.60% (95% CI 87.1–99.9) but reduced specificity to 53.60% (95% CI 33.9–72.5), with an accuracy of 79.70% (95% CI 68.3–88.4) and an LR+ of 2.10 (95% CI 1.41–3.14).

### Hemoglobin

Hb levels showed a moderate diagnostic value, with an AUC of 0.736 (95% CI 0.614–0.859). A threshold of ≤11.45 g/dL yielded a sensitivity of 70.7% (95% CI 54.5–83.9) and specificity of 82.10% (95% CI 63.1–93.9). The PPV was 85.30% (95% CI 68.9–95.0), the NPV was 65.70% (95% CI 47.8–80.9), and the overall accuracy was 75.40% (95% CI 63.5–84.9). The LR + was 3.96 (95% CI 1.75–8.98), and the LR − was 0.36% (95% CI 0.22–0.59).

### White blood cell count

WBC count showed strong diagnostic performance, with an AUC of 0.926 (95% CI 0.865–0.987). A cut-off of ≥10,455 cells/µL yielded a sensitivity of 82.9% (95% CI 67.9–92.8) and specificity of 92.9% (95% CI 76.5–99.1), with a PPV of 94.4% (95% CI 81.3–99.3) and an NPV of 78.8% (95% CI 61.1–91.0). The overall accuracy was 87.0% (95% CI 77.0–94.0), and the LR + was 11.6 (95% CI 3.03–44.5), indicating strong discrimination.

### Erythrocyte sedimentation rate

ESR exhibited moderate diagnostic utility, with an AUC of 0.750 (95% CI 0.628–0.872). At a cut-off of ≥45 mm/hr, sensitivity was 75.0% (95% CI 58.8–87.3), and specificity was 75.0% (95% CI 55.1–89.3). The PPV was 81.1% (95% CI 64.8–92.0), the overall accuracy was 75.0% (95% CI 63.0–84.7), the LR + was 3.00 (95% CI 1.54–5.84), and the LR − was 0.33 (95% CI 0.19–0.59).

### C-reactive protein

CRP showed good diagnostic performance, with an AUC of 0.864 (95% CI 0.774–0.954). A threshold of ≥18 mg/L yielded a sensitivity of 80.0% (95% CI 64.4–90.9) and a specificity of 88.9% (95% CI 70.8–97.6). The PPV was 91.4% (95% CI 76.9–98.2), the NPV was 75.0% (95% CI 56.6–88.5), the overall was 83.6% (95% CI 72.5–91.5), and the LR + was 7.2 (95% CI 2.45–21.2).

### Interleukin-6 (IL-6) level

IL-6 showed moderate discriminative ability for distinguishing active disease from remission, with an AUC of 0.781(95% CI 0.652–0.910). A threshold of ≥16 pg/mL yielded a sensitivity of 59.3% (95% CI 38.8–77.6) and a specificity of 91.7% (95% CI 73.0–99.0). The PPV was 88.9% (95% CI 65.3–98.6), the NPV was 66.7% (95% CI 48.2–82.0), the overall was 74.5% (95% CI 60.4–85.7), and the LR + was 7.11% (95% CI 1.82–27.8).

### Diagnostic performance of combined biomarkers for differentiating active disease from remission in patients with anti-IFN-γ AAbs

The diagnostic utility of combined biomarkers was evaluated for their ability to distinguish active disease from remission in patients with anti-IFN-γ AAbs (Table 4).

### Combination of anti-IFN-γ AAb titers, ESR, CRP, WBC, and IL-6

The combined use of anti-IFN-γ AAb titers, ESR, CRP, WBC count, and IL-6 level exhibited excellent diagnostic performance, with an AUC of 0.903 (95% CI 0.808–0.997). This panel achieved a sensitivity of 88.9% (95% CI 70.8–99.0),

**Table 4. Performance of combined laboratory parameters in predicting disease.**

| Biomarker combination | AUC (95% CI) | Sensitivity % (95% CI) | Specificity % (95% CI) | LR+ (95% CI) | LR− (95% CI) | PPV % (95% CI) | NPV % (95% CI) | Accuracy % (95% CI) |
|---|---|---|---|---|---|---|---|---|
| Anti–IFN-γ AAbs+ESR+CRP+WBC+IL-6 | 0.903 (0.808–0.997) | 88.9 (70.8–97.6) | 91.7 (73.0–99.0) | 10.7 (2.81–40.5) | 0.12 (0.04–0.36) | 92.3 (74.9–99.1) | 88.0 (68.8–97.5) | 90.2 (78.6–96.7) |
| Anti–IFN-γ AAbs+ESR+CRP | 0.869 (0.775–0.964) | 85.0 (70.2–94.3) | 88.9 (70.8–97.6) | 7.65 (2.61–22.4) | 0.17 (0.08–0.36) | 91.9 (78.1–98.3) | 80.0 (61.4–92.3) | 86.6 (76.0–93.7) |
| Anti–IFN-γ AAbs+ESR+CRP+WBC | 0.913 (0.834–0.992) | 90.0 (76.3–97.2) | 92.6 (75.7–99.1) | 12.2 (3.19–46.3) | 0.11 (0.04–0.28) | 94.7 (82.3–99.4) | 86.2 (68.3–96.1) | 91.0 (81.5–96.6) |
| CRP+ESR+WBC | 0.869 (0.775–0.964) | 85.0 (70.2–94.3) | 88.9 (70.8–97.6) | 7.65 (2.61–22.4) | 0.17 (0.08–0.36) | 91.9 (78.1–98.3) | 80.0 (61.4–92.3) | 86.6 (76.0–93.7) |
| CRP+ESR+IL-6 | 0.789 (0.660–0.919) | 70.4 (49.8–86.2) | 87.5 (67.6–97.3) | 5.63 (1.90–16.7) | 0.34 (0.19–0.62) | 86.4 (65.1–97.1) | 72.4 (52.8–87.3) | 78.4 (64.7–88.7) |

Anti-IFN γ Aabs titers; Anti-interferon-gamma autoantibodies titers, AUC; area under the curve, CRP; C-reactive protein, ESR; erythrocyte sedimentary rate, Hb; hemoglobin, IL-6; interleukin 6 level, LR+; positive likelihood ratio, LR-; negative likelihood ratio, NPV; negative predictive value, PPV; positive predictive value, WBC; White blood cell count.

specificity of 91.7% (95% CI 73.0–99.0), PPV of 92.3% (95% CI 74.9–99.1), NPV of 88.0% (95% CI 68.8–97.5), and accuracy of 90.2% (95% CI 78.6–96.7). The likelihood ratios were 10.7 (LR+) (95% CI 2.81–40.5) and 0.12 (LR−) (95% CI 0.04–0.36), confirming strong diagnostic value.

### Combination of anti-IFN-γ AAb titers, ESR, and CRP

The combination of anti-IFN-γ AAb titers, ESR, and CRP yielded an AUC of 0.869 (95% CI 0.775–0.964). This panel achieved a sensitivity of 85.0% (95% CI 70.2–94.3) and specificity of 88.9% (95% CI 70.8–97.6), with a PPV of 91.9% (95% CI 78.1–98.3), NPV of 80.0% (95% CI 61.4–92.3), and accuracy of 86.6% (95% CI 76.0–93.7). The likelihood ratios were 7.65 (LR+) (95% CI 2.61–22.4) and 0.17 (LR−) (95% CI 0.08–0.36), indicating good diagnostic performance.

### Combination of anti-IFN-γ AAb titers, ESR, CRP, and WBC

The combination of anti-IFN-γ AAb titers, ESR, CRP, and WBC showed the highest diagnostic accuracy, with an AUC of 0.913 (95% CI 0.834–0.992). It achieved a sensitivity of 90.0% (95% CI 76.3–97.2) and specificity of 92.6% (95% CI 75.7–99.1), with a PPV of 94.7% (95% CI 82.3–99.4), NPV of 86.2% (95% CI 68.3–96.1), and accuracy of 91.0% (95% CI 81.5–96.6). The likelihood ratios were 12.2 (LR+) (95% CI 3.19–46.3) and 0.11 (LR−) (95% CI 0.04–0.28), indicating strong discriminatory capacity to differentiate between active disease and remission.

### Combination of CRP, ESR, and WBC

The combination of CRP, ESR, and WBC achieved an AUC of 0.869 (95% CI 0.775–0.964), with a sensitivity of 85.00% (95% CI 70.2–94.3) and a specificity of 88.90% (95% CI 70.8–97.6). It yielded a PPV of 91.90% (95% CI 278.1–98.3), an NPV of 80.00% (95% CI 61.4–92.3), and an overall accuracy of 86.60% (95% CI 76.0–93.7). The likelihood ratios were 7.65 (LR+) (95% CI 2.61–22.4) and 0.17 (LR−) (95% CI 0.08–0.36), indicating moderate diagnostic utility.

### Combination of CRP, ESR, and IL-6

The combination of CRP, ESR, and IL-6 yielded an AUC of 0.789 (95% CI 0.660–0.919), with a sensitivity of 70.40% (95% CI 49.8–86.2) and specificity of 87.50% (95% CI 67.6–97.3). It provided a PPV of 86.40% (95% CI 65.1–97.1), NPV of

72.40% (95% CI 52.8–87.3), and overall accuracy of 78.40% (95% CI 64.7–88.7). The likelihood ratios were 5.63 (LR+) (95% CI 1.90–16.7) and 0.34 (LR−) (95% CI 0.19–0.62), indicating moderate diagnostic utility.

## Discussion

The role of anti-IFN-γ AAbs in immunodeficiency and infectious disease, is well established. [19,32] Patients with anti-IFN-γ AAbs frequently experience recurrent and relapsing infection, particularly due to NTM infection and other intracellular organisms. [1,15,16,18] Treatment regimens and duration are determined by the causative pathogens and the extent of organs involvement. [33] Therefore, close monitoring of clinical response and laboratory biomarkers is essential to optimize treatment outcome.

Previous studies reported that serum anti-IFN-γ Aabs titer are a potential biomarker for evaluating disease activity in AIGA patients. anti-IFN-γ Aabs titers fluctuate over the course of the disease, the antibody level decrease when the disease improves with treatment and increase during active disease. [28,29,34] Nithichanon et al. explored the efficacy of anti-IFN-γ AAb titers in distinguishing between patients with active and inactive NTM infection, and demonstrated a significant association between anti-IFN-γ AAb titers,with a cut-off inhibition titer ≥ 1:5,000 and active disease, AUC = 0.8795. [29]

In our study, anti-IFN-γ AAb titers exhibited strong diagnostic performance in differentiating active disease from remission as definition, with an AUC of 0.893. At a cut-off of >1:50,000, the titers demonstrated high specificity (92.90%) and a moderate sensitivity (65.90%), supporting their utility in confirming disease activity. Lowering the cut-off to ≥1:10,000 increased sensitivity but reduced specificity, highlighting the trade-off between sensitivity and specificity. Measurement of anti-IFN-γ AAb titers by inhibitory ELISA may further enhance the diagnostic workflow by directly assessing functional neutralization of IFN-γ. This assay may complement indirect ELISA, particularly in cases with borderline titers or discordant clinical features. Although anti-IFN-γ Aabs titers are a high-performance biomarker reflecting disease status, their use may be limited in some centers due to resource constrains.

Other laboratory biomarkers have been investigated for disease monitoring in Anti-IFN-γ Aabs patients. A prospective cohort study comparing AIGA-positive and AIGA-negative patients with mycobacterium tuberculosis infection reported a significantly highly levels of WBC, neutrophils, monocytes, CRP, ESR, globulin and immunoglobulin G level in AIGA-positive (p < 0.05). [35] Laisuan et al. [26] reported changes in biomarkers among AIGA patients receiving adjunctive immunosuppressive therapy, with remission associated with decreased levels of WBC, ESR and CRP compared with active disease. [26] Similar trends were observed by Angkasekeinai et al. [25], who demonstrated that CRP, ESR and WBC were meaningful biomarkers for categorizing disease activity in patients with anti-IFN-γ Aabs patients.

In this study, additional inflammatory biomarkers including Hb, WBC, ESR, CRP, and IL-6 were evaluated for diagnostic performance. Among individual markers, WBC count exhibited the strongest discriminatory power, with an AUC of 0.926 and a high LR+ of 11.61. WBC counts were significantly elevated during periods of active disease, supporting their utility in monitoring disease activity in patients with anti-IFN-γ AAbs.

CRP and ESR are well-established acute-phase reactants commonly used to monitor inflammation and demonstrated moderate to good diagnostic utility in this study. CRP had an AUC of 0.864, with an optimal cut-off of ≥18 mg/L, achieving good sensitivity (80.00%) and specificity (88.90%). ESR exhibited moderate utility, with an AUC of 0.750. This findings are consistent with the known roles of ESR and CRP levels in reflecting inflammatory responses to infection in patients with anti-IFN-γ Aabs. [26] IL-6, another inflammatory cytokine, also correlated with disease activity demonstrated good specificity with cut off ≥ 16 pg/mL, although to a lesser extent than CRP and ESR, with an AUC of 0.789. These results suggest that IL-6 may serve as a supplementary biomarker but is less reliable than other inflammatory biomarkers in use alone. However, its availability may be limited in routine clinical practice.

Comorbid conditions in AIGA patients, including concurrent infections, anemia, hepatic diseases, renal impairment, and other inflammatory states, may influence biomarker level independently of disease activity. These factors may result in either overestimation or underestimation of biomarker performance when distinguishing active disease from remission.

Nevertheless, the combined use of multiple biomarkers may improve overall diagnostic accuracy in determining disease activity by mitigating the limitations of individual markers.

The combination of biomarkers improved diagnostic performance. The addition of anti-IFN-γ AAb titers and WBC increased both the sensitivity and specificity of the diagnostic models. The combination of anti-IFN-γ AAb titers, ESR, CRP, and WBC count demonstrated excellent performance (AUC = 0.913), with high sensitivity (90.00%), specificity (92.60%), and PPV (94.70%), providing the highest overall diagnostic accuracy among the tested panels. This underscores the benefit of integrating multiple laboratory biomarkers for optimal disease monitoring. Although, the combination of CRP, ESR, and WBC was less accurate (AUC = 0.869), it still exhibited moderate diagnostic performance, suggesting that this simpler panel may serve as a practical and more accessible option for routine clinical monitoring, particularly in resource-limited settings.

Despite the promising results, several limitations should be acknowledged. The retrospective design and relatively small sample size may have limited the statistical power of specific analyses. Additionally, although the biomarkers evaluated provide valuable diagnostic information, they do not fully capture the complex immunological mechanisms underlying anti-IFN-γ AAb-associated immunodeficiency. Prospective studies with larger, more diverse cohorts are needed to validate the clinical utility of these biomarkers. Future research should also examine longitudinal changes in biomarker levels to assess their predictive value for disease recurrence, progression, or flare-ups.

## Conclusion

This study highlights the importance of combining anti-IFN-γ AAb titers with inflammatory markers (ESR, CRP, WBC count, and IL-6) to distinguish active disease from remission in AIGA patients. Combined biomarker panels can provide more reliable and comprehensive diagnostic information than individual markers, offering significant potential for clinical management of AIGA. Further studies are required to refine and validate their utility in routine practice.

## Author contributions

**Conceptualization:** Putthapon Teepapan, Porpon Rotjanapan, Wannada Laisuan.

**Data curation:** Putthapon Teepapan, Apinya Chungcharoenpanich, Kanokkarn Pinyopornpanish, Supa Oncham, Prawat Chantharit, Wannada Laisuan.

**Formal analysis:** Putthapon Teepapan, Wannada Laisuan.

**Investigation:** Putthapon Teepapan.

**Methodology:** Putthapon Teepapan.

**Project administration:** Putthapon Teepapan, Supa Oncham.

**Resources:** Supa Oncham.

**Supervision:** Porpon Rotjanapan, Wannada Laisuan.

**Validation:** Putthapon Teepapan, Apinya Chungcharoenpanich, Kanokkarn Pinyopornpanish, Prawat Chantharit, Wannada Laisuan.

**Writing – original draft:** Putthapon Teepapan, Wannada Laisuan.

**Writing – review & editing:** Putthapon Teepapan, Apinya Chungcharoenpanich, Kanokkarn Pinyopornpanish, Prawat Chantharit, Porpon Rotjanapan, Wannada Laisuan.

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
