## [Decision Letter · Decision Letter 0]

28 Dec 2025

**ACADEMIC EDITOR: Major revision**

We look forward to receiving your revised manuscript.

Kind regards,

Marwan Salih Al-Nimer, MD, PhD

Academic Editor

PLOS One

Journal Requirements:

2. Please include your tables as part of your main manuscript and remove the individual files. Please note that supplementary tables (should remain/ be uploaded) as separate "supporting information" files.’

“This study received grant from Ramathibodi Hospital, Mahidol university”

5. We note that your Data Availability Statement is currently as follows: All relevant data are within the manuscript and its Supporting Information files.

7. Please upload a copy of Figure 1, to which you refer in your text on page 9. If the figure is no longer to be included as part of the submission please remove all reference to it within the text.

8. Please include a copy of Table 4 which you refer to in your text on page 4.

**Additional Editor Comments:**

This manuscript is not well prepared. It needs a major revision

1: Revise the introduction and supplement it with updated references.

2: Methods are not clear. Rewrite the methods, specify the eligibility, inclusion, exclusion, the parametes.......etc.

3; The results were without tables and figures???????

4: Rewrite the discussion. The authors added only one reference to the discussion????

Reviewers' comments:

Reviewer's Responses to Questions

**Comments to the Author**

1. Is the manuscript technically sound, and do the data support the conclusions?

Reviewer #1: Partly

Reviewer #2: Partly

2. Has the statistical analysis been performed appropriately and rigorously?

Reviewer #1: No

Reviewer #2: Yes

3. Have the authors made all data underlying the findings in their manuscript fully available?

Reviewer #1: No

Reviewer #2: No

4. Is the manuscript presented in an intelligible fashion and written in standard English?

Reviewer #1: Yes

Reviewer #2: Yes

Reviewer #1: The analysis isn´t enough descriptive. Previous comorbidity pathologic conditions doesn´t mention if earliest disease manifestations can be related to the so called initial adult-onset immunodeficiency nor if there are previous treatments achieved during early comorbidities, evolution or manifestations of disease. Analysis doesn´t mention changes for expresión of activation markers among immunocompetent population nor if there is a shift from a Th1/Th2, inflammatory/anti-inflammatory profile of population, and even in the secretion of cytokine profiles of subpopulations. It would be appropriate to know if such immunologic adult-onset disease parameters are related to clinical course changes from active to remission of disease. There is also a lack of other inflammatory and/or fibrin- or coagulation-related factors, such as ferritin, D-dimer, PLT and thrombin-related times which would perpetuate inflammatory profiles. In my opinión, there are some issues that remain to be defined in a context-dependent manner as in maintaining active treatment in order to avoid progression after ending treatment. In summary changes in remission states have their own positive clinical characteristics at remission contrary to those at initial adult-onset diagnosis but it means to check clinical course of disease along time in order to maintain anti-inflammatory changes.

Reviewer #2: In this manuscript, the authors investigate the correlation between anti–interferon‑γ autoantibody titers measured by inhibitory ELISA and clinical disease activity, as well as inflammatory biomarkers in adult‑onset immunodeficiency. The topic is clinically important, especially given the high prevalence of anti‑IFN‑γ autoantibody–associated immunodeficiency in Southeast Asia and the ongoing need for reliable biomarkers for disease monitoring. The study is generally well-written and contains meaningful data; however, several major and minor issues should be addressed before the manuscript can be considered for publication.

Major Comments

1. The manuscript repeatedly refers to Table 1–4 and Figure 1, but these materials do not appear in the provided document. This significantly impairs the ability to evaluate data presentation, statistical validity, interpretation of ROC curves and diagnostic performance. Therefore, please ensure that all tables and figures referenced in the text are included, properly formatted, and placed within the main manuscript, or uploaded as separate supporting files.

2. The study is described as retrospective, nested within a prospective cohort, but essential methodological details require clarification. Were samples collected and analyzed at uniform time points for all patients? Were the biomarkers measured at the same visit as disease activity assessment? These clarifications are needed to ensure internal validity and avoid bias.

3. Inclusion and exclusion criteria need further explanation. Whether all consecutive patients with anti‑IFN‑γ auto Abs were included. Whether disease activity distribution (40 active vs 29 remission samples) might reflect selection bias. Clarification is required to validate the generalizability of the results.

4. There is no information on duration of illness, prior antimicrobial or immunosuppressive therapy, timing of relapse/remission cycles. These variables could significantly influence biomarkers (e.g., CRP, ESR, IL‑6) and may confound interpretation of disease activity. Therefore, this information should be included.

5. The Methods section requires further clarification and expansion, particularly regarding Inhibitory ELISA methodology, cytokine assays and immune cell population analysis. At present, essential methodological details are missing, which limits the reproducibility and interpretability of the results.

6. The manuscript reports AUC, cutoffs, sensitivity, specificity, etc., but confidence intervals (CIs) are missing for many key metrics. CIs are essential for interpreting robustness and reproducibility. Logistic regression analysis is briefly mentioned, but the regression model structure, variable selection strategy, model diagnostics (multicollinearity, goodness‑of‑fit), are insufficiently described.

7. The study concludes that combining biomarkers improves diagnostic performance; however, there is limited discussion on clinical feasibility, particularly since IL‑6 testing may not be routinely available. The impact of comorbidities—e.g., infections, anemia, hepatic or renal dysfunction—on CRP, ESR, ALP, or WBC levels should be acknowledged. How these factors may confound biomarker interpretation should be discussed

8. Discussion should address clinical implementation challenges

Minor Comments

1. The table on comorbidities is only described in text. Once tables are added, ensure demographic and clinical characteristics are clearly presented.

2. Some thresholds (e.g., IL‑6 >16) should indicate units explicitly (pg/mL). Make units consistent throughout.

3. The discussion is generally sound, but could be strengthened by comparing predictive performance with prior studies more explicitly and adding comments on how inhibitory ELISA may complement or replace indirect ELISA in clinical practice.

**Do you want your identity to be public for this peer review?** For information about this choice, including consent withdrawal, please see our Privacy Policy

Reviewer #1: **Yes:** Luis Molto Delgado

Reviewer #2: No

---

## [Author Response · Author response to Decision Letter 1]

9 Feb 2026

We thank the Editor and Reviewers for your careful evaluation of our manuscript and for your constructive and insightful comments. We have addressed each comment point by point and revised the manuscript accordingly. All changes have been incorporated into the revised version, and we believe these revisions have strengthened the clarity, scientific rigor, and clinical relevance of the manuscript. We hope that the revised manuscript now meets the expectations of the Editor and Reviewers, and we appreciate the opportunity to revise our work.

---

## [Editor Report · Decision Letter 1]

15 Feb 2026

Dear Dr. Laisuan,

Thank you for submitting your manuscript to PLOS ONE. After careful consideration, we feel that it has merit but does not fully meet PLOS ONE’s publication criteria as it currently stands. Therefore, we invite you to submit a revised version of the manuscript that addresses the points raised during the review process.

**ACADEMIC EDITOR: Minor revision**

We look forward to receiving your revised manuscript.

Kind regards,

Marwan Salih Al-Nimer, MD, PhD

Academic Editor

PLOS One

Journal Requirements:

Additional Editor Comments:

Retype the references according to the PLoS ONE guideline

---

## [Author Response · Author response to Decision Letter 2]

16 Feb 2026

Dear Editor

The revised manuscript had been update as reviewer comments

- Reference were updated as recommended and had been checked

- Updated cover letter with financial statement as attached

-Upload name of files were applied as recommendation

---

## [Editor Report · Decision Letter 2]

17 Feb 2026

Correlation between clinical disease activity and interferon-Υ autoantibody titers measured by inhibitory ELISA, and inflammatory biomarkers in adult-onset immunodeficiency associated with anti-interferon-Υ autoantibodies

PONE-D-25-58456R2

Wannada Laisuan

Dear Dr. Wannada Laisuan,

We’re pleased to inform you that your manuscript has been judged scientifically suitable for publication and will be formally accepted for publication once it meets all outstanding technical requirements.

Kind regards,

Marwan Salih Al-Nimer, MD, PhD

Academic Editor

PLOS One
---

## [Editor Report · Acceptance letter]

PONE-D-25-58456R2

PLOS One

Dear Dr. Laisuan,

I'm pleased to inform you that your manuscript has been deemed suitable for publication in PLOS One. Congratulations! Your manuscript is now being handed over to our production team.

Kind regards,

on behalf of

Professor Marwan Salih Al-Nimer

Academic Editor

PLOS One